# Development of Composite Microbial Products for Managing Pine Wilt Disease in Infected Wood Stumps

**DOI:** 10.3390/microorganisms12122621

**Published:** 2024-12-18

**Authors:** Yanzhi Yuan, Yanna Wang, Yong Li, Laifa Wang, Lu Yu, Jian Hu, Xiangchen Cheng, Shan Han, Xizhuo Wang

**Affiliations:** 1Key Laboratory of Forest Protection of National Forestry and Grassland Administration, Ecology and Nature Conservation Institute, Chinese Academy of Forestry, No. 2 Dongxiaofu, Haidian, Beijing 100091, China; yyz41742@163.com (Y.Y.); lylx@caf.ac.cn (Y.L.); nema@caf.ac.cn (L.W.); 18304017749@163.com (L.Y.); miatabest67@gmail.com (J.H.); 2Chinese Society of Forestry, Chinese Academy of Forestry, No. 2 Dongxiaofu, Haidian, Beijing 100091, China; wangyanna@csf.org.cn; 3Center for Biological Disaster Prevention and Control, National Forestry and Grassland Administration, Shenyang 110034, China; chengxiangchen1986@163.com; 4College of Forestry, Sichuan Agricultural University, Chengdu 611130, China; hanshan6618@163.com

**Keywords:** *Bursaphelenchus xylophilus*, pine wilt disease, wood-decay fungi, microbial products, wood stumps

## Abstract

Wood-decay fungi, including white- and brown-decay fungi, are well known for their ability to degrade lignin and cellulose, respectively. The combined use of these fungi can increase the decomposition of woody substrates. Research has indicated that these fungi also exhibit inhibitory effects against *Bursaphelenchus xylophilus*, the causative agent of pine wilt disease (PWD). In this study, we investigated a composite microbial formulation that efficiently decomposes pine wood while inhibiting *B. xylophilus*. We initially established a correlation between the degradation rate of wood blocks and fungal biomass, underscoring the necessity of optimizing biomass for effective treatment. A systematic approach involving a one-way test, a Plackett–Burman design, a steepest ascent experiment, and a Box–Behnken design, was employed to optimize the fermentation process. Validation trials were conducted in a 10-L fermenter. The bioagent’s efficacy and safety were assessed through field applications in a forest, with a focus on wood degradation capacity and *B. xylophilus* mortality rate. Additionally, the environmental impact of the microbial products was evaluated by analysing soil quality around treated areas to ensure that the formulation did not adversely affect soil health.

## 1. Introduction

Pine wilt disease (PWD), caused by *Bursaphelenchus xylophilus*, is a globally important disease in forests and is classified as a quarantine disease in several countries. It originated in North America and poses a serious threat to forests in East Asia, especially in Japan [1], China [2], Korea [3], and Europe [4,5]. Controlling the spread of infectious diseases has become extremely important in recent decades, and immediate measures must be taken to contain their spread. The key to these efforts for PWD is the management of stumps, which are the main reservoirs of the disease. Stumps are important sources of infestation, and pest treatment with netting and pesticides is costly, however, biological treatment is a possible route for managing the disease [6]. In this context, wood-decay fungi have emerged as effective biological control products. Their enzymatic capacity allows them to break down woody substrates while inhibiting the growth and reproduction of *B. xylophilus*, which is responsible for PWD [7,8]. Field applications have shown that certain fungi, including white-, brown- and soft-decay fungi, have different effects on the degradation of lignin, cellulose and hemicellulose and the control of *B. xylophilus* [9,10,11]. Notably, white-decay fungi are able to completely mineralize lignin, and [12], brown-decay fungi mainly target cellulose and hemicellulose [13], whereas soft-decay fungi alter the structure of lignin but do not degrade it completely.

Recent studies on microbial inoculation have emphasized the use of monoculture wood decay fungi, particularly white-decay fungi, which breaks down complex structures, thereby enhancing the degradation process [14]. However, in natural settings, the degradation of lignocellulose involves diverse fungal species that contribute to this process [13]. Compared with monocultures, coculture systems of fungi, such as a combination of the white-decay fungus *Bjerkandera adusta* and the brown-decay fungus *Gloeophyllum sepiarium*, have demonstrated improved efficiency in woody material degradation; this is achieved through an increase in α-galactosidase activity and an enhancement in the enzymatic degradation of wood [15]. Similarly, cocultures with different types of white-decay fungi presented increased lignocellulose enzyme activity [16].

Liquid fermentation has been demonstrated to be effective for cultivating a broad spectrum of microorganisms. This method promotes rapid microbial growth due to the even distribution of nutrients within the liquid medium. The current research on liquid fermentation is focused on refining the culture medium and fermentation conditions to maximize microbial activity and productivity. Various experimental approaches, including traditional one-factor experiments, Plackett–Burman designs, and response surface methodologies, have been implemented to optimize these parameters [17,18,19]. These improvements in microbial cultivation and fungal coculture are pivotal in enhancing the biodegradation processes essential in environmental management and industrial applications.

In this study, we aimed to increase the biomass of a composite fungal culture by correlating the mycelial growth of *Lenzites betulinus* (a white-decay fungus) and *Fomitopsis pinicola* (a brown-decay fungus), both of which were prescreened in our laboratory, with the degradation rates of wood blocks. We adopted a liquid shake flask culture method as our primary culturing technique and established a foundational fermentation protocol. We subsequently utilized a one-way test to determine which culture conditions significantly impacted production. We then employed a Plackett–Burman design to identify critical factors and utilized the Box–Behnken response surface methodology to model these factors, explore their interactions, and determine optimal conditions. Upon finalizing our culture conditions, we created composite microbial products and applied them in a forest setting. We evaluated the effectiveness of these products by monitoring changes in the populations of *B. xylophilus* and *Monochamus alternatus*, as well as the degradation rates of treated wood stumps before and after inoculation. Additionally, to confirm the environmental safety of our agent, we analysed soil samples from around the treated areas to evaluate changes in soil fertility before and after the application of the agent. This comprehensive approach allowed us to optimize and validate the effectiveness of our biodegradation strategy in a controlled yet naturalistic environment.

## 2. Materials and Methods

### 2.1. Biological Materials

The highly pathogenic *B. xylophilus* isolate QH-1 [20], isolated from infected *Pinus koraiensis* (Liaoning, China) was cultured for a fortnight on *Botrytis cinerea* fungi grown on barley [21]. The mixed phase of *B. xylophilus* was extracted at room temperature via the Baermann funnel technique [22] and then washed with Sterile water for the experiments. *B. xylophilus* was counted according to Wang’s method [23].

Two wood-decay fungi, *Lenzites betulinus* LB-01 (CFCC NO. 57600) and *Fomitopsis pinicola* FP-09 (CFCC NO. 80995), were used for the study. They were provided by the China Forestry Culture Collection Center.

The fungus was subsequently grown on potato dextrose agar (PDA) plate culture media (200 g of potato (Beijing, China), 20 g of glucose (Beijing Solarbio Science & Technology Co., Ltd., Beijing, China), 20 g of agar (Xilong Scientific Co., Ltd., Shantou, China), per L), stored at 4 °C, inoculated with a 0.5 cm^2^ piece of the fungus and incubated at 25 ± 1 °C until mycelia covered the entire surface of the plates.

### 2.2. General Information About the Test Site

The test site was located in the southeastern Shanxi Province, at the upper reaches of the Han River, which is in the northern subtropical zone with a continental humid monsoon climate. The sunshine duration is moderate, and the average annual temperature is 15.7 °C. Rainfall is abundant, with an average annual precipitation of 799.3 mm, and the climate is mild. The test site was set up in Huxin village, Yinghu town, Hanbin District, Ankang city, at an altitude of 467.27 m (±10.0 m). For the forest test, we selected stumps of *Pinus massoniana* that were infected with PWN (Pine Wood Nematode). 

### 2.3. Effects of Coculture Strains on B. xylophilus

Two wood-decay fungi were grown on potato dextrose agar (PDA) media for 7 days at 28 °C. Each starter inoculum was inoculated into a 250-mL conical flask containing 150 mL of fermentation medium (g/L: 3% corn flour (Beijing, China), 1.5% peptone (Beijing Aoboxing Bio-tech Co., Ltd., Beijing, China), 0.05% MgSO_4_·7H_2_O (Xilong Scientific Co., Ltd.), and 0.2% KH_2_PO_4_ (Xilong Scientific Co., Ltd.), initial pH 5) [6]. The flasks were subjected to constant temperature shaking at 28 °C and 160 rpm.

The fermented fungal mixture was coated onto a PDA plate, and when the strains grew across the entire plate, 2000 strips of *B. xylophilus* [24] were added to each dish. *B. cinerea* strain was used as a positive control strain, with 10-dish replicates for each treatment. The test strain QH-1 and the cocultured fungi were incubated in a constant-temperature incubator at 25 °C for 8 and 13 days, respectively. At the end of the incubation period, 5 dishes each of the test strain QH-1 and the cocultured fungi were taken out, and the *B. xylophilus* in the plates were separated by the Bellman funnel method. The numbers of *B. xylophilus* adults and larvae on the plates were counted separately [9] to analyse the effects of inoculation of strips with different time on the reproduction of *B. xylophilus* (Table 1).

### 2.4. Relationship Between the Mycelial Biomass and Mass Loss of Wood

The experimental method was based on the relevant standard with modifications [25]. Fresh *P. massoniana* woodblocks were cut into 20 mm × 20 mm × 5 mm chips. Prior to inoculation, the woodblocks were placed in a 105 °C oven until a constant weight was achieved, and the weight of the woodblocks was measured before the test. The aseptic woodblocks were placed in water for 12 h and then autoclaved at 121 °C for 30 min. A total of 40, 60, 80, 100, or 120 μL of fermentation broth (monoculture or coculture) was added to 90 mm petri dishes containing PDA, and after four days, the above woodblocks were inoculated (Figure 1a). After inoculation, the cultures were placed in a constant-temperature incubator at 28 °C for 30 days. After 30 days, surface mycelia and impurities were gently scraped off. The woodblocks were oven-dried at 105 °C until a constant weight was reached [26]. The weight of the woodblocks after the test was measured, and the percentage mass loss of the woodblocks was calculated via Equation (1):(1)$$ML\left(Mass\;loss\;percentage\right)=\frac{W_{1}-W_{2}\;}{W_{1}}\times 100\%$$

*W*_1_: Oven-dry weight of the woodblock sample before the test; 

*W*_2_: Oven-dry weight of the woodblock sample after the test.

**Figure 1 microorganisms-12-02621-f001:**
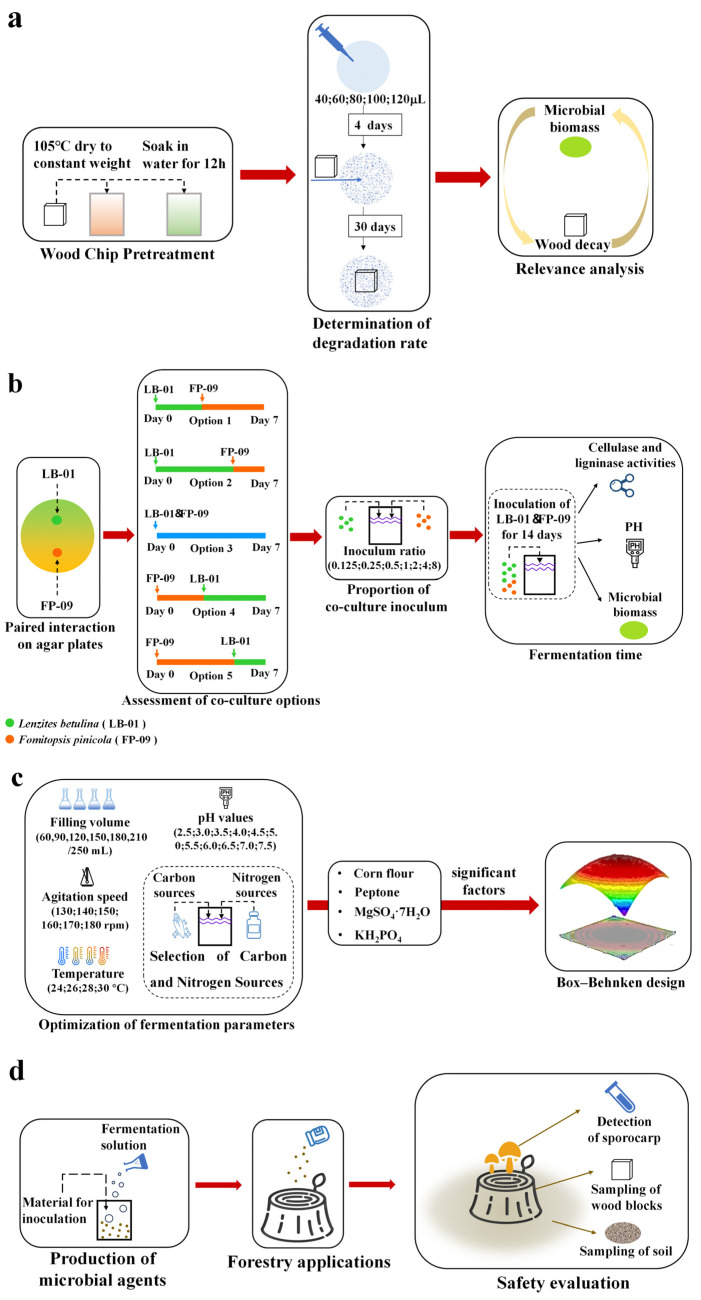
Experimental flowchart of (**a**) the relationship between mycelial biomass and wood decay; (**b**) assessment of coculture options; (**c**) establishment of a cocultivation optimization model; and (**d**) forestry applications.

### 2.5. Assessment of Coculture Options

#### 2.5.1. Direct Inhibition Test

Preliminary coculture experiments were conducted to establish the absence of significant inhibition between LB-01 and FP-09. Three replicates of each group of treatments were used, for a total of three treatments. Five-millimetre-diameter agar discs were cut from actively growing colonies, and the LB-01 and FP-09 combination agar discs were placed mycelium-side down on Petri dishes containing PDA medium [27] at a distance of 4.5 cm. Control experiments involving the same strains were also performed. The cultures were incubated at 25 °C in a constant temperature and humidity incubator. The diameter of the colony facing the other colony was measured from the agar side of the Petri dish via a straight-edged ruler every other day from the first day and recorded on days 1, 3, 5, and 7. The inhibition rate was calculated via the following formula [28]:(2)$$I[inhibition\;percentage]=(\frac{R1[colony\;radius\;in\;control]-R2[colony\;radius\;in\;test]}{R1\;})\times 100$$

#### 2.5.2. Inoculation Sequence

There were five options for the cocultivation of LB-01 and FP-09 (Figure 1b), with 3 replicates for each option. In option 1, FP-09 was inoculated on day 3 after LB-01 was grown in a 250-mL flask. In option 2, FP-09 was inoculated on day 5 after LB-01 was grown in a 250-mL flask. In option 3, LB-01 and FP-09 were inoculated simultaneously at the beginning of flask fermentation. In option 4, LB-01 was inoculated on day 3 after FP-09 was grown in a 250-mL flask. In option 5, LB-01 was inoculated on day 5 after FP-09 was grown in a 250-mL flask. The temperature was maintained at 28 °C, and the shaker speed was controlled at 160 rpm with constant temperature shaking. Samples were taken after 7 days. The assessment of each cocultivation option was based on the biomass in the flask [29].

#### 2.5.3. Inoculum Ratio

To determine the optimum inoculation ratio, LB-01 and FP-09 were inoculated in 250-mL triangular flasks containing 150 mL of 3% corn flour, 1.5% peptone, 0.05% MgSO_4_·7H_2_O, and 0.2% KH_2_PO_4_, initial pH 5 at different ratios (0.125, 0.25, 0.5, 1, 2, 4, and 8). The experiment consisted of a repeated measures design with three treatments, each replicated 3 times. The flasks were subjected to constant-temperature shaking at 28 °C and 160 rpm for 7 days. The optimal vaccination ratio was based on the biomass in the flask. The optimal inoculation ratio for the subsequent experiments was determined on the basis of these evaluations.

#### 2.5.4. Fermentation Time

To determine the optimal coculture time, activated mycelial clusters with 5 mm side lengths on LB-01 and FP-09 agar plates were attached to conical flasks at a 2:1 ratio, and monocultures were used as controls. These clusters were then transferred to 250-mL triangular flasks containing 150 mL of seed medium. The flasks were shaken at 28 °C and 160 rpm for 14 days [30]. Three replicate samples were taken every 2 days to measure the biomass, Pondus Hydrogenii and enzyme activity during fermentation. On the basis of these tests, the optimal coculture time was determined for the following experiments. 

#### 2.5.5. Biomass Evaluation

The cocultivated biomass of the two strains of wood-decay fungi was evaluated via the mycelial dry weight method. Constant-weight filter paper was used for filtering the fermentation broth. After the fermentation broth was filtered, the filter paper and mycelia were dried at 80 °C for 10 h. The mycelial biomass was determined by weighing the filter paper and mycelia [31]. 

#### 2.5.6. Enzyme Activity Assays

The samples for enzymatic measurements were aseptically withdrawn from the culture broth every two days (1 mL) and stored at −80 °C in plastic sample tubes. The samples were rapidly thawed, vortexed, and lysed (13,000× *g*, 4 °C, 5 min) prior to analysis [32]. The cellulase enzyme activity was determined via the DNS method with reference to the relevant specification [33]. Laccase activity was determined with reference to the ABTS method [34]. The lignin peroxidase assay was modified from the method of Tien [35]. First, 0.5 mL of 200 mM tartaric acid buffer was added to a cuvette with a volume of 1.4 mL, and 0.1 mL of 40 mM linalool was added. X μL of the enzyme mixture to be tested was added (X = 0~100), and purified water (400−X) was added. The water bath temperature was maintained at 30 °C, and the reaction was initiated by adding 0.01 mL of a 20 mM H_2_O_2_ solution. The absorbance at 310 nm was determined quickly and measured once more after 1 min, and the difference between the two was used to calculate the change in absorbance per minute (OD310 change) [36]. The manganese peroxidase assay was modified from methods by Wariishi [37]. First, 0.5 mL of 100 mM malonate-sodium malonate buffer was added to a cuvette with a volume of 1.4 mL, and 0.1 mL of 10 mM MnSO_4_ solution was added. The enzyme mixture to be tested was X μL (X = 0~50), and pure water (400−X) μL was added. In a 30 °C water bath, 0.01 mL of 10 mM H_2_O_2_ solution was added to start the reaction, the absorbance at 270 nm was quickly determined, the absorbance was determined again after 1 min, and the difference between the two was calculated, which was the change in absorbance per minute (OD270 change) [36]. 

### 2.6. Establishment of a Coculture Optimization Model

#### 2.6.1. Selection of Carbon and Nitrogen Sources

To determine the optimal carbon and nitrogen sources, the effects of various carbon sources (glucose, sucrose, maltose, and corn flour) and nitrogen sources (peptone, soybean powder, yeast powder, and NH_4_NO_3_) on the coculture of wood-decay fungi in fermentation medium were compared. All possible combinations of carbon and nitrogen sources were tested, resulting in fermentation media were compared. All possible combinations of carbon and nitrogen sources were tested, resulting in 16 different compounds (Table 2), each with a C:N ratio of 2:1. For each combination, both the corn flour and the soybean powder were sieved through a 40-mesh screen. The final compound selection was based on mycelial biomass [38]. All the experiments were performed in quintuplicate, and the results are presented as the means ± standard deviations.

#### 2.6.2. Selection of Other Fermentation Conditions

Optimization was performed for five additional key factors, including pH (2.5, 3.0, 3.5, 4.0, 4.5, 5.0, 5.5, 6.0, 6.5, 7.0, and 7.5), temperature (24, 26, 28, and 30 °C), rotational speed (130, 140, 150, 160, 170, and 180 rpm), and filling volume (60, 90, 120, 150, 180, and 210/250 mL) (Figure 1c). The inoculum ratio refers to the ratio of the biomass density (g·L^−1^) of LB-01 to that of FP-09. The total inoculum densities of LB-01 and FP-09 were kept at approximately 0.02 g·L^−1^ [39]. All the experiments were performed in quintuplicate, and the results are presented as the means ± standard deviations. 

#### 2.6.3. Plackett–Burman Design

An 8-factor, 12-run Plackett–Burman design experiment was conducted (Table 3). The selection of low (−1) and high (1) levels for some factors (Table 3) was based on reports in previous literature. Five replicates per run were performed when the experiments were conducted. A total of 60 flasks were randomly assigned to 12 runs. For statistical modelling, the first-order polynomial linear model shown in Equation (3) was employed:(3)$$Y=\beta_{o}\sum \beta_{n}X_{n}\left(n=1,2,\ldots ,k\right)$$
where *Y* is the predicted mycelial biomass (g), *β*_0_ is the model intercept, *β_n_* is the linear coefficient and *X_n_* is the coded level of the independent variable. The main effects plot for the Plackett–Burman design was obtained via “Design-Expert 13” software. The significance level was set to a *p* value < 0.05. Significant independent variables were considered for further optimization via a Box–Behnken design. Because not all conditions are essential for fermentation, nonsignificant conditions could be removed. All the experiments were performed in quintuplicate, and the results are presented as the means ± standard deviations.

#### 2.6.4. Path of Steepest Ascent Method

The response surface fitting equation was established according to the optimal level factor approach. The optimum level scope of the key factors was examined according to the path of the steepest ascent method. The ascending direction of the steepest ascent method (Table 4) was determined by the positive and negative effects of the key factors according to the results of the Plackett–Burman design [31]. All the experiments were performed in quintuplicate, and the results are presented as the means ± standard deviations.

#### 2.6.5. Box–Behnken Design

The key factors considered included peptone (%), KH_2_PO_4_ (%) and temperature (°C). A Box–Behnken design with 3 factors in one block with 17 runs (Table 5) was employed for the study. The design consisted of the three factors (peptone, KH_2_PO_4_ and temperature) at three equidistant levels (4%, 5% and 6% for peptone; 0%, 0.2% and 0.4% for KH_2_PO_4_; and 19, 22 and 25 °C for temperature). In the optimization experiments, mycelial biomass (Y) was recorded as the response (dependent) variable. The different components were mixed at predetermined concentrations in 250-mL flasks and placed in a shaking incubator at different temperatures for a specified period. The designed model was further validated via random combinations of the independent variables. All the results were analysed by employing “Design-Expert 13” software to determine the optimum fermentation conditions [40]. All the experiments were performed in quintuplicate, and the results are presented as the means ± standard deviations.

Five groups of parallel experiments were conducted under the optimal conditions to analyse whether the actual results were within the 95% confidence interval of the predicted values to verify the accuracy of the model prediction [31].

#### 2.6.6. Fermentation at the 10-L fermenter Level

In this study, a 10-L fermenter was used to produce fungi under optimized fermentation conditions. With the loading volume at 7 L, the secondary seeds were inoculated into the fermenter at 10% inoculum. The aeration rate during fermentation was 1 L·min^−1^, the rotational speed was 100 L·min^−1^ in the early stage, and the parameters were adjusted according to the foam output in the later stage. A 2-L aseptic reflux flask was used to control the rate of foam reflux, and the rate of foam flow into the reflux flask was adjusted to maintain the volume of fermentation broth at 7 L. After fermentation, samples were taken to determine biomass [38].

### 2.7. Production and Application of Coculture Microbial Products

We demonstrated the ability of cocultured fungi to inhibit the growth and reproduction of *B. xylophilus* and accelerate the degradation of wood blocks. Moreover, we correlated degradation efficacy with mycelial biomass, achieving biomass maximization in the short term. If fermentation broth can be transformed into microbial products that can be applied to forests, it will provide a theoretical and practical basis for the control of pine wilt disease leading to epidemics. Chen [6] used five different wood-decay fungi as microbial products to inoculate the stumps of *P. massoniana Lamb*, and the mass loss of stump samples was greater than that of control samples after 70 days. Therefore, we adopted the formulation and inoculation method of Chen [6] with appropriate improvements. We inoculated the stumps with cocultured fungi and conducted a detailed survey of factors such as *M. alternatus* oviposition grooves and feathering holes on the stump surface, and the number of *B. xylophilus* within the stumps. *B. xylophilus* are able to enter the respiratory system of *M. alternatus* and parasitize in its body. When the feathered *M. alternatus* feeds on healthy *Pinus*, *B. xylophilus* enters the interior of the tree through the wounds, and jeopardizes the growth of *Pinus*. Adults usually lay their eggs on weakened wood, so if there are a certain number of *B. xylophilus* in the stumps and *M. alternatus* have successfully feathered, there is a great possibility of secondary transmission of pine wilt disease. Therefore, it is necessary to investigate the factors on the surface and inside of the stumps. Meanwhile, changes in soil quality around the stumps were the focus of our investigation because microbial products can only be truly valuable if they are harmless to the surrounding environment (Figure 1d).

#### 2.7.1. Preparation of Microbial Products

A total of 250 mL of fermentation broth was injected into a bag containing inoculation materials (wood chips 100 g, glucose 4 g, corn flour 4 g, peptone 1 g, KH_2_PO_4_ 0.6 g, MgSO_4_·7H_2_O 0.3 g, pectin 4 g (Shanghai Macklin Biochemical Technology Co., Ltd., Shanghai, China), and L-Glutamic acid monosodium salt hydrate 4 g (Shanghai Macklin Biochemical Technology Co., Ltd.)) sterilized at 121 °C and then mixed homogeneously. Seven days later, a small amount of mycelium was inoculated into PDA media to test the growth conditions, and the growth conditions were considered good if the mycelia grew well. If the mycelia grew well, a forest inoculation test was carried out [6]. 

#### 2.7.2. Inoculation of Stumps

Stumps with diameter greater than 10 cm and no signs of decomposition in the xylem were selected. The surfaces of the stumps were cut with a chainsaw and then evenly coated with a prepared agent (80 g of microbial product on a stump with a diameter of 10 centimetres and 15 g of agent for each additional centimetre increase in diameter) and then covered with a breathable film. Samples (wood blocks measuring 2 cm × 2 cm × 1 cm) were taken before the test and after 180 days, and a control was set up. The samples were divided into two parts: one part was placed in an oven at 100 ± 5 °C to dry to a constant weight, with each wood block weighed to measure the mass loss percentage, and the other part was used for the isolation and counting of *B. xylophilus* [9]. All the experiments were performed in quintuplicate, and the results are presented as the means ± standard deviations.

#### 2.7.3. Isolation and Identification of Fungal Sporocarps

A tissue isolation method was used. The collected sporocarps of LB-01 and FP-09 were surface-sterilized with 75% alcohol and 1% mercuric acid. The fungal mass was torn open, and the inner soybean grain-sized tissues were placed on PDA plates, cultured at a constant temperature of 25 °C and protected from light. The mycelium was transferred several times to obtain pure cultures, and PDA slants were used for seed preservation.

Small amounts of the genomic DNA of strains LB-01 and FP-09 was extracted via liquid nitrogen lance tip milling in EP (Eppendorf) tubes. The genomic DNA was adjusted to the appropriate concentration with sterilized ddH_2_O and stored at −20 °C in a refrigerator for use. The ITS sequences of strains LB-01 and FP-09 were amplified via primers for ITS4/ITS5 (5′-TCCTCCGCTTATTGATATATGC-3′/5′-GGAAGTAAAAGTCGTAACAAGG-3′). The amplification system was 50 μL and contained 1.5 μL of DNA template, 2.0 μL of each upstream and downstream primer, 25 μL of 2 × Es Taq MasterMix (Dye) and 19.5 μL of ddH_2_O. The PCR amplification conditions were as follows: predenaturation for 3 min at 94 °C, denaturation for 30 s at 94 °C, annealing for 30 s at 55 °C and extension for 30 s at 72 °C for 30 cycles. The amplified products were verified by agarose gel electrophoresis and sent to the Beijing Genomics Institute for sequencing. The sequenced sequences were removed from the vector and primer sequences. Then, homologous comparison was performed via the BLAST tool, and the ITS sequences of related published strains were downloaded. Phylogenetic analysis was performed via ME-GA11, and a phylogenetic tree was constructed via the Kimura2-parameter model via the neighbour–joining (NJ) method. A total of 1000 bootstrap analyses (bootstrapping) were performed [41]. 

#### 2.7.4. Safety Assessment

Before and after microbial agent inoculation, 300 g of soil samples were collected from the 10–20 cm soil layer around the logging stakes via a 5-point sampling method with a ring knife. The apomictic material on the ground surface was removed before sampling with the ring knife. The soil was mixed into one sample, placed in a self-sealing bag and transported to the laboratory at low temperature for the determination of soil physicochemical properties. Cation exchange capacity (CEC) was analysed according to methods by Helling [42]. The total nitrogen (TN) content was determined via the Kjeldahl method [43]. The total phosphorus (TP) content was determined via methods by Sommers [44]. The total potassium (TK) content was determined via the method of Nathan Gammon [45]. The soil ECe was measured at a 1:5 soil-to-water ratio via a conductivity meter [46]. Acid phosphatase (EC 3.1.3.2, orthophosphoric-monoester phosphohydrolase, acid optimum) activities were determined according to methods by Tabataba [47].

To comprehensively assess the effects of agent application on the soil around the stumps, a fuzzy integrated evaluation method was used to evaluate soil fertility around the stumps before and after inoculation. Principal component analysis was used to derive the common factor variance and calculate the weights of each index. The degree of affiliation of each indicator was calculated via the affiliation function. S-type functions were used for the above six indicators [48]. After combining the weights and affiliation values of each indicator, the integrated fertility index (*IFI*) of the soil was calculated via Equation (4):(4)$$IFI=\sum_{i=1}^{n}\left(W_{i}\times F_{i}\right)$$
where *n* is the number of participating indicators, and *W_i_* and *F_i_* are the weight and affiliation value of the *i*th indicator, respectively.

### 2.8. Statistical Analysis

Differences between treatment groups were compared using the One-way analysis of variance (ANOVA) test, followed by Tukey’s honestly significant difference (HSD) test using IBM SPSS Statistics 26 with significance set at *p* < 0.05.

## 3. Results

### 3.1. Inhibition of B. xylophilus by Coculture Strains

After maturation of the cocultured fungi on PDA plates, 2000 *B. xylophilus* were collected and cocultured for 8 and 13 days. The mean number of *B. xylophilus* counted per dish isolate was statistically analysed, and the significance of the difference was determined (Table 1). On day 8, the number of *B. xylophilus* cultured in *B. cinerea* increased approximately 50-fold (100,187), with a larval proportion of 43%. The number of *B. xylophilus* isolated from the cocultured colony was 94.7 per dish on average, of which 28.7 were larvae, accounting for 30.3%, indicating that the cocultured colony had an inhibitory effect on the growth of *B. xylophilus* adults and the reproduction of larvae. The number of *B. xylophilus* isolated from *B. cinerea* after 13 days increased approximately 55.5-fold (111,098.7), and the percentage of larvae increased to 44.5%. The number of *B. xylophilus* isolated from the coculture colonies decreased to an average of 24.7 per dish, and no *B. xylophilus* larvae were isolated, suggesting that the inhibitory effect of the cocultured colonies on *B. xylophilus* increased over time, with a more potent effect on larvae. The average number of adults and larvae per dish of cocultured colonies differed significantly (≤0.05). 

### 3.2. Correlation Between the Mycelial Biomass and Mass Loss of Wood

A bivariate Pearson’s test revealed a significant positive correlation between mass loss percentage and mycelial biomass (r = 0.977, *p* = 0.000 < 0.05). The normality test of the mass loss percentage (Y) of mycelial biomass yielded a statistic of 0.948 and a significance level of Sig. = 0.494 > 0.05, indicating that the dependent variable, mass loss percentage (Y), was a normally distributed variable and could be subjected to regression analysis. The regression equation was established as Y = 29.341 + 0.172X; as the mycelial biomass (X) increased by 1 µL, the mass loss percentage increased by 0.172%.

### 3.3. Mycelial Interaction Between Fungi Under Coculture

The intraspecies and interspecies interactions of white- and brown-decay fungi were evaluated on Petri dishes to clarify the effects of their interactions. In petri dishes inoculated with the same fungi (LB-01 vs. LB-01, FP-09 vs. FP-09), mycelia from two opposing colonies merged in the central region of the plate on days 5 and 3 of incubation (Figure 2a). The results showed that mycelium from the same colony was not disturbed, or growth was not inhibited, indicating the absence of intraspecific competition. However, some competitive interactions were observed in the cocultures of the two fungi (LB-01 and FP-09). During a 7-day incubation, LB-01 colonized faster than FP-09 did, hindering the growth of its competitor through mycelial interference [13]. A fence of high mycelial density was also observed on day 5 when the two fungal hyphae were in contact. On the plates inoculated with LB-01 and FP-09, the inhibition of LB-01 was negative on the third day (Figure 2b), suggesting that FP-09 promoted the growth of LB-01. The inhibition of FP-09 reached a maximum value of 22.52% on the fifth day and then started to decrease. The inhibition was close to zero on the seventh day at 1.43%, suggesting that there was almost no inhibition between the two species at the late stage of incubation. The results show that white- and brown-decay fungi can be grown together in the same medium, which implies that sequential inoculation of white and brown-decay is feasible. Under coinoculation, we can expect both fungi to grow higher than they do in their corresponding monocultures, which is due to their interactions.

### 3.4. Assessment of Inoculation Options

To determine the optimal coculture options, LB-01 and FP-09 were inoculated into fermentation media according to the selected experimental design (Figure 1b). As shown in Figure 2c, the mycelial biomasses of the cocultured microorganisms reached 1.29, 1.62, 1.72, 1.51 and 1.49 g in Options 1, 2, 3, 4, and 5, respectively. The mycelial biomass was greater in Option 3 than in the other options. Therefore, regarding biomass production, Option 3 was determined to be an optimal choice [29]. 

The effects of the LB-01:FP-09 inoculation ratio on mycelial biomass are shown in Figure 2d. The mycelial biomass reached a maximum value (1.68 g) when the inoculation ratio was 2:1. The results of the experiment revealed that, within a certain range, the greater the proportion of LB-01 was than that of FP-09, the greater the mycelial biomass. LB-01 grew faster than FP-09 did in the antagonism experiments, which was also confirmed during liquid fermentation, during which the two strains were able to maximize their coculture growth at a ratio of 2:1.

### 3.5. Selection of the Optimal Fermentation Time for Coculture

The cocultured strains entered the logarithmic growth phase on the sixth day of incubation in the value-added medium (Figure 3a), the number of cells increased rapidly, and the pH decreased sharply, peaked on the 10th day (Figure 3b), and then entered the receding phase in which the pH increased continuously. LB-01 entered the logarithmic phase at the beginning of the incubation, and the pH decreased continuously, peaked on the 10th day, and then entered the receding phase. FP-09 grew slowly in the value-added medium. The pH decreased linearly from 5.2 to 2.6 on the 2nd day and then fluctuated within a 0.4 unit range of 2.6 and below (Figure 3b). During the value-added phase, low pH, low nutrient concentrations and high harmful metabolite concentrations caused by fungal growth inhibited bacterial growth. The increase in pH during the decline phase may be related to the autolysis of bacterial cells. In the late logarithmic phase, the bacterial concentration was highest, along with the viability and fertility of the cells. Therefore, 10 days was determined to be the most appropriate fermentation time for coculture [30].

The laccase activity of LB-01 was the highest and reached a maximum of 244.61 U·L^−1^ on day 4. The cocultured strains had relatively high laccase activity with LB-01, and the trend remained the same up to day 12. The laccase activity of FP-09 was consistently low relative to that of LB-01. This occurred mainly because laccase plays an important role in the degradation of lignin, whereas brown rot fungi do not have the ability to degrade lignin independently (Figure 3c). The lignin peroxidase activity of FP-09 consistently tended to decrease from day 4 to the end of fermentation. The lignin peroxidase activity of the coculture and LB-01 cultures changed in opposite directions until day 12, reaching a minimum value of 0.30 U·L^−1^ and a maximum value of 1.40 U·L^−1^ on day 2, respectively (Figure 3d). The manganese peroxidase activity of the coculture system fluctuated more widely, reaching a maximum value of 13.25 U·L^−1^ on day 4. The manganese peroxidase activity of the monoculture tended to increase, and the trends were not synchronous, reaching values of 14.29 U·L^−1^ and 8.95 U·L^−1^ on day 14 (Figure 3e). The cellulase activity was greater in both the cocultures and monocultures. The coculture and FP-09 reached the maximum value on day 6, and the trends in both remained the same until day 10, which may be related to the fact that the cellulase activity in the preculture period was driven mainly by brown rot fungi. However, the cellulase activity of white rot fungi reached 73.59 U·L^−1^ on day 10, and the maximum activity of brown rot fungi was only 65.82 U·L^−1^ (Figure 3f), suggesting that white rot fungi are also promising in terms of cellulase activity [32].

### 3.6. Optimized Design and Experiment for Culture Conditions

The different combinations of carbon and nitrogen sources resulted in significant differences on the growth of the cocultured strains (Figure 4a). The maximum mycelial biomass (2.20 g) and a high density of mycelial spheres were observed when the carbon source was corn flour and the nitrogen source was peptone, thus, corn flour and peptone were determined to be the most suitable carbon and nitrogen sources. The best mycelial growth was observed at pH 5 (Figure 4b), which indicated that LB-01 and FP-09 grew best under neutral to acidic conditions. Mycelial biomass increased with increasing loading volume (Figure 4c), which occurred because the amount of nutrients increased with increasing loading volume. The slope (biomass/fill volume) decreased at a fill volume of 150 mL, indicating that 150 mL of medium per 250 triangular flask was an appropriate ratio for the effective space. A rotational speed that is too fast will result in failure to form mycelia, and a speed that is too slow will result in insufficient oxygen in the triangular flasks. It was found that 160 rpm was the rotational speed at which mycelial biomass was maximized (Figure 4d). As shown in Figure 4e, 28 °C was the optimum temperature for the mycelial growth of LB-01 and FP-09.

The Plackett–Burman design (PBD) methodology was employed with eight factors as variables: corn flour, peptone, MgSO_4_·7H_2_O, KH_2_PO_4_, pH, temperature, filling capacity, and rotational speed. Because the mycelial biomass produced by the original culture conditions was only 2.09 g, it was necessary to screen the culture conditions for the production of more mycelia. The key factors associated with the PBD experiment were screened, and the mycelial biomass data are shown in Table 3. After a first-order polynomial model was fitted, the coefficient of determination (R^2^) was determined to be 0.9835 (Table 3), which indicated a good fit. The four factors (filling capacity, temperature, peptone, and KH_2_PO_4_) had more significant effects on mycelial biomass than did the original conditions (Table 6).

Temperature, peptone, and KH_2_PO_4_ were chosen as important factors for subsequent experiments because filling capacity is limited by the loading capacity of the vessel in actual production. The filling capacity was fixed at 150 mL per flask. Corn flour, MgSO_4_·7H_2_O, pH, and rotational speed were not significant factors in the Placket–Burman design experiment, and the experimental conditions were kept the same as the initial experimental conditions.

To further evaluate the comprehensive effects of temperature, peptone, and KH_2_PO_4_ on mycelial biomass, the steepest ascent method was used to determine the centre point of the response surface. It was evident (Table 4) that the optimal mycelial biomass reached the highest peak at the fifth step, at which the temperature and peptone and KH_2_PO_4_ contents had values of 22 °C, 5%, and 0.2%, respectively.

To determine the interactive effects of temperature, peptone and KH_2_PO_4_, a Box–Behnken design was used to estimate the interactive effects of the three factors on mycelial growth. Mycelial biomass was obtained at the centre point (run 5) (Table 5), and a significant model (*p* < 0.0001) with a nonsignificant lack of fit (*p* = 0.0885) and an R^2^ value of 0.9721 was obtained (Table 7). Therefore, the model adequately fit the experimental data.


$$\begin{array}{cc} \mathrm{Mycelial}\;\mathrm{biomass}= & -47.3179+2.65555\;\mathrm{temperature}+7.763847\;\mathrm{peptone}+10.90876{\mathrm{KH}}_{2}{\mathrm{PO}}_{4}+0.01078\;\mathrm{temperature}\times \mathrm{peptone}\\ & -0.2647\;\mathrm{temperature}\times {\mathrm{KH}}_{2}{\mathrm{PO}}_{4}+0.91708\;\mathrm{peptone}\times {\mathrm{KH}}_{2}{\mathrm{PO}}_{4}-0.05738\;{\mathrm{temperature}}^{2}-0.814175\;{\mathrm{peptone}}^{2}-\\ & \;\;\;23.6127{\mathrm{KH}}_{2}{{\mathrm{PO}}_{4}}^{2} \end{array}$$


The overall effects were determined by plotting two factors as the independent variables and mycelial biomass as the dependent variable, with the three factors fixed at their optimum levels (Figure 4f–h). The model revealed highly significant (*p* < 0.0001) interactions of peptone × peptone, KH_2_PO_4_ × KH_2_PO_4_ and other interactions, indicating that the response values of the factors are not simple linear relationships but rather partial quadratic relationships [31].

### 3.7. Optimal Culture Conditions and Validation

According to the Box–Behnken design experiment, the mycelial biomass was predicted to be 4.05 g under optimum conditions (temperature 23.1527 °C, peptone 5.03271%, and KH_2_PO_4_ 0.198949%, as shown in Figure 5; for ease of implementation, the three values were considered to be 23.2 °C, 5%, and 0.2%, respectively) (Figure 5). The experimental results under the optimum conditions revealed a mycelial biomass of 4.02 g, which was close to and thus validated the predicted result from the BBD experiment. The actual measured yield was close to the predicted yield, indicating that the established model was in agreement with the actual outcomes, therefore, optimization of the fermentation conditions of the strain for maximum mycelial biomass was effective and feasible.

### 3.8. Scaled-Up Experiments in a 10-L Fermenter

Mycelial biomass was determined in a 10-L fermenter under optimized and initial culture conditions. The initial biomass was 13.821 g·L^−1^, and the optimized biomass reached a value of 30.531 g·L^−1^. The optimized biomass was 2.21 times greater than the yield in the initial medium under 10-L fermentation conditions. The biomass at the end of fermentation in the 10-L fermenter was 1.14 times greater than that at the end of shaking fermentation, indicating that the fungi were able to fully utilize the nutrients in the medium when they were cultured in the fermenter.

### 3.9. Application and Evaluation of Microbial Products

A survey of stumps infected by PWD is shown in Table 8. The number of oviposition grooves made by *M. alternatus* was related to the height of the stumps. There were essentially no grooves on shorter stumps, but when the height of the stumps was greater than 20 cm, the possibility of grooves was greater. The grooves of *M. alternatus* can also appear in windswept wood in forests, which is often hollow or dead. The number of oviposition grooves increased after 180 days in the uninoculated stumps. The inoculated stumps remained unchanged, suggesting that the microbial products effectively inhibited *M. alternatus* from laying eggs on the stumps. We also observed that the number of *M. alternatus* did not increase after inoculation, whereas the number of *M. alternatus* increased on uninoculated stumps.

The number of *B. xylophilus* isolated from the inside of diseased stumps ranged from 23.4 to 669.4 (Table 8), with an average of 314.3 per stump. The number of *B. xylophilus* decreased in all cases after 180 days. The number of *B. xylophilus* in inoculated stumps was 0, indicating that the microbial products had a 100% inhibitory effect on *B. xylophilus*. The uninoculated stumps still contained *B. xylophilus* in numbers ranging from 0 to 200. The degradation of stumps by microbial products was more pronounced (Table 8). The average degradation rate of inoculated stumps was 50.28%, which was 3.12 times greater (16.1%) than that of uninoculated stumps. Microbial product application resulted in faster xylem degradation, greater decay, and a reduced *B. xylophilus* population, which was related to the parasitic characteristics of the organism. Therefore, the microbial products were able to directly and indirectly inhibit the growth and reproduction of *B. xylophilus*.

To confirm that LB-01 and FP-09 had effects on *B. xylophilus* and stumps, we collected two kinds of sporocarps growing on the surface of stumps for molecular biological isolation and identification. After PCR amplification and sequencing, the ITS sequences of LB-01 and FP-09 were obtained with a length of 500 bp (Figure 6a). An online BLAST comparison via the NCBI website revealed a sequence with 98.88% similarity to *L. betuilnus*. The ITS sequences of *Lentinus swartzii*, *Lentinus bertieri*, *Lentinus badius*, *Lentinus sajor-caju* and *Lenzites betulinus* were downloaded from the ITS sequences of published strains of related *Lenzites* genera and *Cerrena unicolor*. Phylogenetic analysis via MEGA11 revealed that strain CFCC NO. 57600 clustered with *Lenzites betulinus* with 100% support (Figure 6c). These results combined with the morphological characteristics of the strain indicated that CFCC NO. 57600 was *Lenzites betulinus*. Another sequence showed 99.57% similarity to *F. pinicola*. We downloaded the ITS sequences of *Fomitopsis cf. meliae*, *Fomitopsis meliae*, *Fomitopsis pinicola*, *Fomitopsis nivosa*, *Fomitopsis palustris*, and other published strains of related *Fomitopsis* and *Aspergillus niger*. Phylogenetic analyses were performed via MEGA11. The ITS sequences of CFCC NO. 80995 were clustered with those of *Fomitopsis pinicola*, and the support rate was 100% (Figure 6b). These results combined with the morphological characteristics of the strain indicated that CFCC NO. 80995 was *Fomitopsis pinicola*.

After the effectiveness of the cocultured microbial products was confirmed, the effect of the microbial products on the surrounding organisms and soil became the focus of our investigation. First, we examined shrubs and trees within 500 m of the inoculated stumps and found that there were no corresponding sporocarps attached to these plants, which is related to the parasitic characteristics of wood-decay fungi. Wood-decay fungi mainly parasitize dead wood and maintain growth by decomposing lignin and cellulose. Therefore, the microbial products have little effect on living plants.

Soil samples were obtained from 10–20 cm around the stumps for analysis via a five-point sampling method. Significance analyses of the soil samples before, after, and without microbial product application revealed no significant differences (Figure 7). Compared with those in the preinoculation period, the soil total nitrogen, total phosphorus, and total potassium contents slightly increased (Figure 7b–d), indicating that the application of the microbial products promoted elemental cycling in the soil to a certain extent. The acid phosphatase content decreased from 27.88 U·g^−1^ (before) to 21.24 U·g^−1^ (after); the value for the control group was 19.57 U·g^−1^ (Figure 7f). The overall decrease in acid phosphatase content after 180 days indicated that the intensity of phosphorus biotransformation in the soil decreased, however, this decrease was not due to inoculation with the microbial products but rather due to normal metabolic processes in the soil. Cation exchange capacity and electrical conductance essentially remained unchanged (Figure 7a,e). The weights of the indicators were determined via principal component analysis, as shown in Table 9. The turning point values were determined on the basis of the relevant literature, as shown in Table 10. The composite values of soil fertility before and after inoculation are shown in Table 11; these values were 0.512, 0.551, and 0.584, respectively, and the differences were not significant. Therefore, the use of microbial products did not significantly affect the quality of the soil around the stumps.

## 4. Discussion

The use of wood-decay fungi for the management of PWD represents a paradigm shift from conventional chemical treatments to more sustainable, environmentally friendly methods. Wood-decay fungi have considerable potential in this context because of their ability to degrade lignin and cellulose, which are integral components of wood. This biodegradation capability not only aids these fungi in the decomposition of infested wood but also in the suppression of *B. xylophilus*, the nematode responsible for PWD.

### 4.1. Integrative Biological Control Approaches

Wood-decay fungi, particularly white- and brown-decay fungi, are recognized for their ability to degrade lignin and cellulose, offering significant potential for biodegradation and pretreatment applications. Their enzymatic systems allow for the breakdown of complex plant cell wall components, which is critical in environmental management strategies, for example, in the degradation of *B. xylophilus*-infected wood stumps [14,49]. Compared with a monoculture, the combined use of these fungi, particularly in a coculture, has been shown to increase the activities of enzymes, such as endoglucanase and laccase, leading to more efficient degradation processes [50,51]. In this study, we aimed to develop composite microbial products using a coculture of *L. betulinus* (a white-decay fungus) and *F. pinicola* (a brown-decay fungus) prescreened in our laboratory to treat PWN-infested wood stumps. This study confirmed that, compared with monocultures, a coculture of white- and brown-decay fungi resulted in superior wood degradation rates. This finding aligns with previous studies that reported enhanced enzymatic activities in fungal cocultures [15]. The coculture of these two fungal types resulted in greater cellulose degradation and increased reducing sugar content, which are critical for the breakdown of woody substrates [52]. Moreover, our results demonstrated that the degradation rate of wood blocks was directly correlated with fungal biomass, a finding supported by statistical analyses such as the bivariate Pearson test. This correlation highlights the importance of maximizing biomass production for effective biodegradation.

### 4.2. Optimization of Fungal Biomass Production

In this study, the rate of degradation of wood blocks by a cocultured microbial mixture was superior to that of monocultures, and the degradation rate and mycelial biomass were used as a bridge to connect the two. Then, the focus of the study was shifted to the maximization of biomass. A key factor for the success of microbial coculture is the inoculation technique. According to previous studies, a sequential inoculation strategy is crucial for the coculture of wood-decay fungi. Although the two fungi selected for this study showed weak mutual inhibition, the complexity of the coculture made it necessary to determine the inoculation order first. Adjusting the inoculation ratio is also an optional method for balancing the lifespan of a coculture. Notably, in this study, the culture process was monitored after the inoculation protocol was established. Contrary to expectation, some connections between biomass and changes in medium pH were found. The changes in enzyme activities were more complex, with the cellulase and ligninase systems showing opposite trends, and the changes in laccase activity were closely linked to the trends in biomass during liquid fermentation.

Fermentation media can provide the nutrients required for the growth and metabolism of fungi, therefore, the composition of a medium strongly affects the growth of mycelia of particular strains and the yield and quality of fermentation products, in which the carbon source, nitrogen source, inorganic salts, and growth factors are referred to as the four major nutrients. Moreover, microbial fermentation conditions are particularly important for the fermentation process. The synthesis of specific metabolites was monitored in real time by monitoring microbial changes. The core factors for the coculture of the 2 wood-decay fungi were loading volume, temperature, peptone, and K_2_PO_4_, as determined through one-factor univariate design, Plackett–Burman design, and response surface methodology optimization techniques; the loading volume was removed because the loading volume is affected by the properties of the container in actual production. A Box–Behnken design (BBD) was used, and mycelial biomass was used as a predictor of optimal coculture conditions. The experimental results under optimal conditions were in general agreement with the predictions of the BBD. The two wood-decay fungi were placed in a 10-L fermenter to amplify the fermentation conditions before and after optimization, and samples were taken to determine the optimization effect.

### 4.3. Environmental Impact and Safety

An essential component of using microbial products in natural settings is ensuring that they do not adversely affect the surrounding environment. In this study, we extensively analysed the potential ecological impacts of introducing these fungal species into forest ecosystems. Soil samples collected from treated areas showed no significant change in soil fertility, indicating that the application of the microbial product is safe and does not disrupt soil microbial dynamics or nutrient availability [17]. Moreover, the environmental assessments conducted before and after application helped elucidate the nontoxic nature of the fungal treatment. Ensuring the ecological safety of biodegradation agents is crucial, as it supports the broader adoption and implementation of biocontrol strategies in forestry management practices. 

### 4.4. Scaling Up and Field Applications

Scaling up the production of fungal biomass without compromising its biodegradation efficiency during the transition from laboratory settings to real-world applications poses various challenges. The use of a 10-L fermenter in this study served as an intermediate step towards larger-scale applications. The results obtained from fermenter-level experiments were promising, showing significant increases in biomass production and enzymatic activity, which are vital for the practical application of microbial products in forest settings. Field trials conducted as part of this study further validated the laboratory findings: treated wood stumps presented a relatively high degradation rate, and the populations of both *B. xylophilus* and *M. alternatus* were reduced to zero. These outcomes underscore the potential of fungal biodegradation agents as viable alternatives to chemical treatments, offering a more sustainable and environmentally friendly approach to managing PWD.

### 4.5. Future Prospects and Challenges

Although the results of this study are encouraging, scaling up this technology to field-level applications will require addressing several challenges, including the consistency of fungal agent production, long-term stability of the treatment, and adaptability to different environmental conditions. Future research should be focused on refining the application methods to increase the practicality and effectiveness of the fungal treatments under diverse forest conditions. Additionally, conducting ongoing monitoring and evaluation will be crucial for understanding the long-term impacts of these fungal treatments on forest health and soil ecology. Collaborative efforts among researchers, forest managers, and policy-makers will be essential to integrate these biological control strategies into existing forest management practices, thereby ensuring a sustainable approach to combating PWD and preserving forest ecosystems.

In conclusion, the use of composite microbial products composed of wood−decay fungi offers a promising alternative to chemical treatments for managing pine wilt disease. By harnessing the natural biodegradation capabilities of these fungi, coupled with strategic enhancements through coculture and optimization techniques, this approach not only addresses the immediate challenges posed by PWD but also contributes to the broader goals of sustainable forestry and environmental conservation.

## Figures and Tables

**Figure 2 microorganisms-12-02621-f002:**
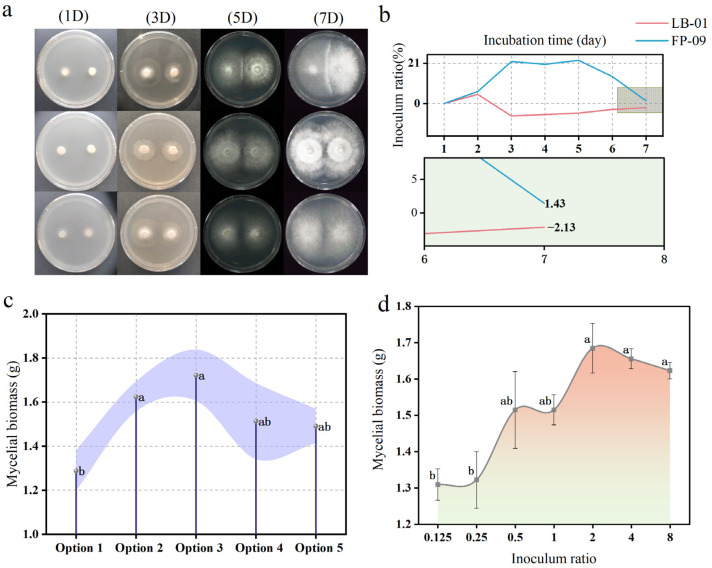
(**a**) Paired interaction on agar plates; (**b**) inhibition rate; (**c**) assessment of coculture options (Option 1: inoculation of LB-01 on Day 0 and inoculation of FP-09 on Day 3; Option 2: inoculation of LB-01 on Day 0 and inoculation of FP-09 on Day 5; Option 3: simultaneous inoculation of LB-01 and FP-09 on Day 0; Option 4: inoculation of FP-09 on Day 0 and inoculation of LB-01 on Day 3; Option 5: inoculation of FP-09 on Day 0 and inoculation of LB-01 on Day 5); (**d**) inoculum ratio. Different lowercase letters represent significant differences.

**Figure 3 microorganisms-12-02621-f003:**
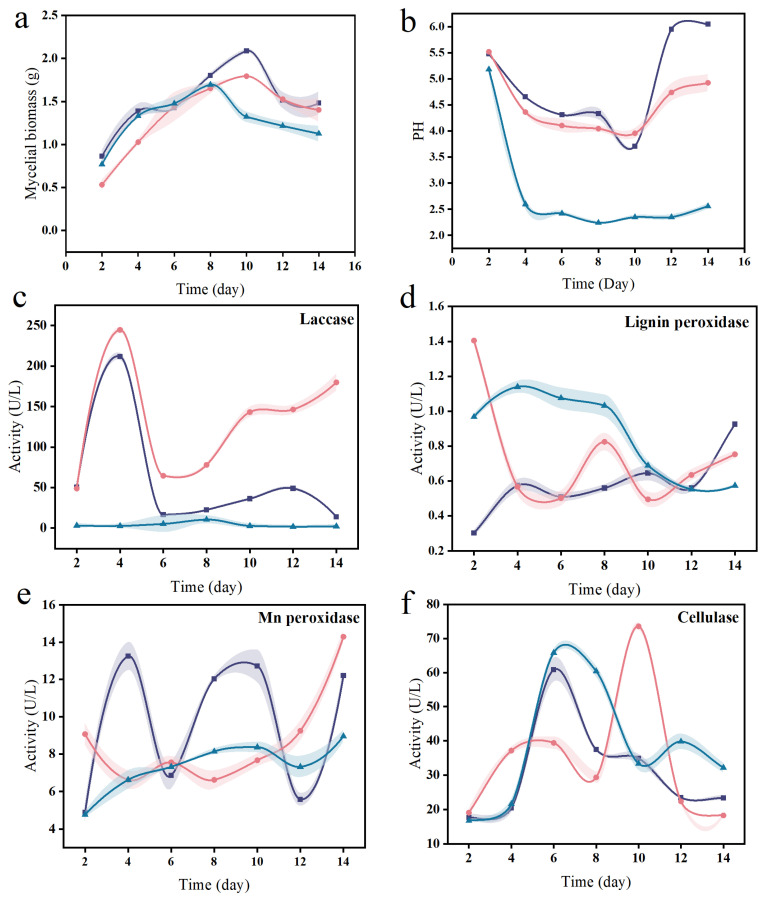
Time course of LB-01 and FP-09 cells. Changes in (**a**) dry weight of LB-01 and FP-09; (**b**) pH; (**c**) laccase activity; (**d**) lignin peroxidase activity; (**e**) Mn peroxidase activity; and (**f**) cellulase activity. LB-01 (red line with spheres), FP-09 (blue line with triangles), and cocultures of LB01 and FP-09 (purple line with squares).

**Figure 4 microorganisms-12-02621-f004:**
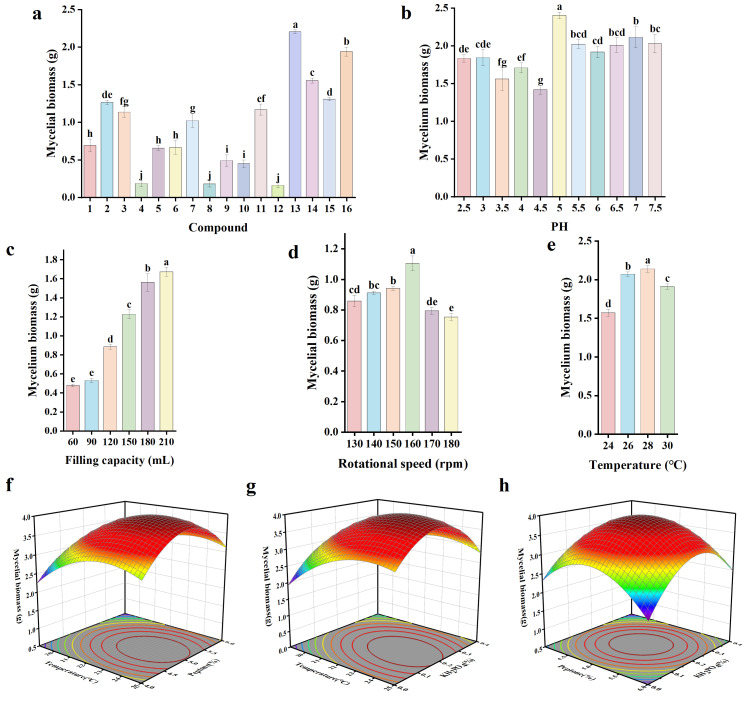
(**a**) Effects of carbon and nitrogen sources on microorganism growth; (**b**) effects of pH on microorganism growth; (**c**) effects of filling volume on microorganism growth; (**d**) effects of agitation speed on microorganism growth; (**e**) effects of temperature on microorganism growth; (**f**) response surface plots of temperature and peptone; (**g**) response surface plots of temperature and KH_2_PO_4_; (**h**) response surface plots of peptone and KH_2_PO_4_. Different lowercase letters represent significant differences.

**Figure 5 microorganisms-12-02621-f005:**
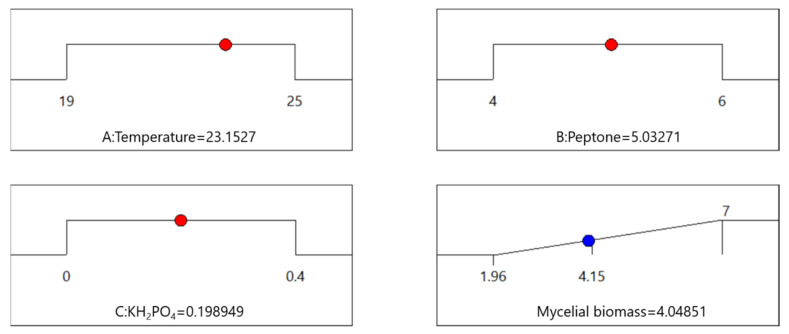
Optimal culture conditions for three significant factors selected by Box–Behnken.

**Figure 6 microorganisms-12-02621-f006:**
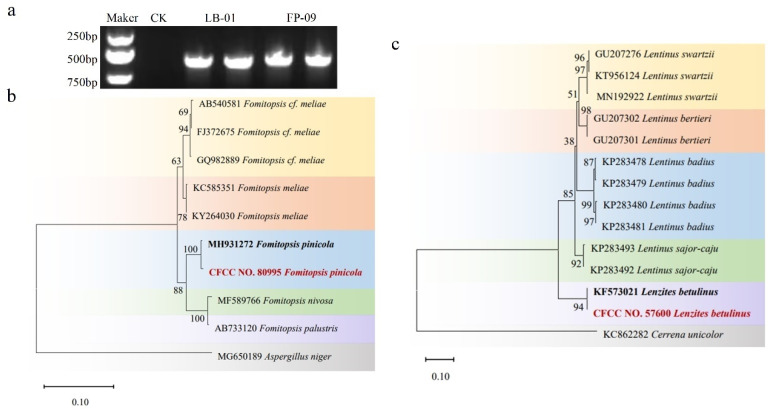
Identification of strains of LB-01 and FP-09: (**a**) electrophoresis analysis of the ITS sequence; (**b**) phylogenetic tree of FP-09; (**c**) phylogenetic tree of LB-01.

**Figure 7 microorganisms-12-02621-f007:**
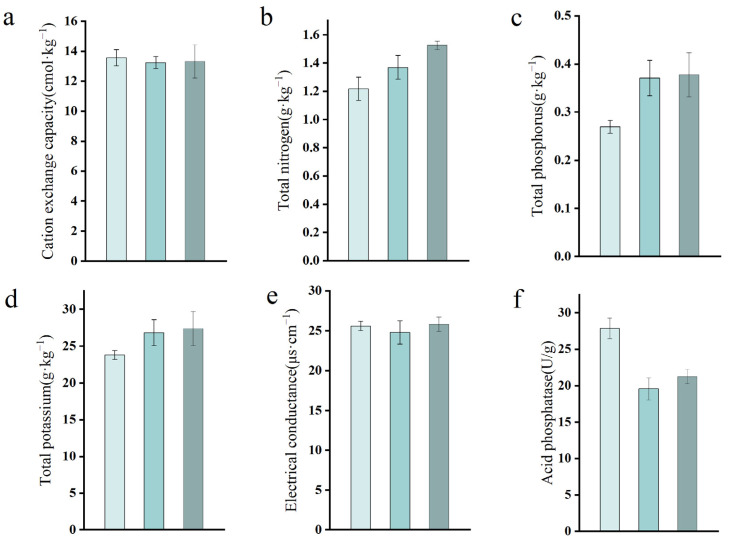
Effects of agents on soil nutrients: (**a**) cation exchange capacity; (**b**) total nitrogen; (**c**) total phosphorus; (**d**) total potassium; (**e**) electrical conductance; and (**f**) acid phosphatase. Colours from lightest to darkest denote pretreatment and noninoculation, respectively.

**Table 1 microorganisms-12-02621-t001:** Changes in the abundance of *B. xylophilus* on LB-01 & FP-09 and *B. cinerea*.

	Time (Day)	*B. xylophilus* Number per Dish		
		Adult	Larva	Larva Ratio (%)
LB-01 & FP-09	8	66.0 ± 14.4 a	28.7 ± 6.0 b	30.3
	13	24.7 ± 9.6 b	0.0 ± 0.0 c	0.0
*B. cinerea*	8	57,079.7 ± 941.3 b	43,107.3 ± 768.8 d	43.0
	13	61,681.7 ± 1060.9 a	49,417 ± 850.5 c	44.5

Different lowercase letters indicate significant differences between the treatments according to Tukey’s HSD test.

**Table 2 microorganisms-12-02621-t002:** Compounds from 16 different carbon sources and nitrogen sources.

Number	Compound	Number	Compound
1	Glucose + Peptone	2	Glucose + Soybean meal
3	Glucose + Yeast	4	Glucose + NH_4_NO_3_
5	Fructose + Peptone	6	Fructose + Soybean meal
7	Fructose + Yeast	8	Fructose + NH_4_NO_3_
9	Maltose + Peptone	10	Maltose + Soybean meal
11	Maltose + Yeast	12	Maltose + NH_4_NO_3_
13	Corn flour + Peptone	14	Corn flour + Soybean meal
15	Corn flour + Yeast	16	Corn flour + NH_4_NO_3_

**Table 3 microorganisms-12-02621-t003:** Mycelial biomasses for 12 runs with 8 factors in the Plackett–Burman design.

Run	Corn Flour	Peptone	MgSO_4_·7H_2_O	KH_2_PO_4_	PH	Temperature	Filling Capacity	Rotational Speed	Mycelial Biomass
	(%)	(%)	(%)	(%)		(°C)	(mL)	(rpm)	(g)
1	2.5 (−1)	1.0 (−1)	0.05 (−1)	0.1 (−1)	4 (−1)	26 (−1)	120 (−1)	150 (−1)	1.19 ± 0.07 ef
2	2.5 (−1)	2.0 (1)	0.15 (1)	0.3 (1)	4 (−1)	26 (−1)	120 (−1)	170 (1)	1.33 ± 0.1 de
3	3.5 (1)	2.0 (1)	0.05 (−1)	0.3 (1)	6 (1)	30 (1)	120 (−1)	150 (−1)	1.13 ± 0.03 f
4	3.5 (1)	2.0 (1)	0.15 (1)	0.1 (−1)	4 (−1)	26 (−1)	180 (1)	150 (−1)	3.52 ± 0.1 a
5	2.5 (−1)	1.0 (−1)	0.05 (−1)	0.3 (1)	4 (−1)	30 (1)	180 (1)	150 (−1)	1.34 ± 0.09 de
6	3.5 (1)	1.0 (−1)	0.15 (1)	0.3 (1)	6 (1)	26 (−1)	120 (−1)	150 (−1)	1.48 ± 0.05 d
7	3.5 (1)	2.0 (1)	0.05 (−1)	0.1 (−1)	4 (−1)	30 (1)	120 (−1)	170 (1)	1.43 ± 0.08 d
8	2.5 (−1)	1.0 (−1)	0.15 (1)	0.1 (−1)	6 (1)	30 (1)	120 (−1)	170 (1)	1.15 ± 0.12 ef
9	2.5 (−1)	2.0 (1)	0.05 (−1)	0.3 (1)	6 (1)	26 (−1)	180 (1)	170 (1)	2.66 ± 0.14 b
10	3.5 (1)	1.0 (−1)	0.15 (1)	0.3 (1)	4 (−1)	30 (1)	180 (1)	170 (1)	1.73 ± 0.14 c
11	3.5 (1)	1.0 (−1)	0.05 (−1)	0.1 (−1)	6 (1)	26 (−1)	180 (1)	170 (1)	2.61 ± 0.03 b
12	2.5 (−1)	2.0 (1)	0.1 5(1)	0.1 (−1)	6 (1)	30 (1)	180 (1)	150 (−1)	2.59 ± 0.02 b

The values for the 8 factors are shown as real values (coded levels). Mycelial biomass is shown as the mean (standard deviation) (*n* = 5). The selection of low (−1) and high (1) levels for some factors was based on previous literature. Different lowercase letters indicate significant differences between the treatments according to Tukey’s HSD test.

**Table 4 microorganisms-12-02621-t004:** Steep climbing test design and results.

Run	Temperature	Peptone	KH_2_PO_4_	Mycelial Biomass
	(°C)	(%)	(%)	(g)
1	34	1	1	1.65 ± 0.04 e
2	31	2	0.8	1.71 ± 0.04 e
3	28	3	0.6	2.16 ± 0.09 d
4	25	4	0.4	2.37 ± 0.03 c
**5**	**22**	**5**	**0.2**	**3.95 ± 0.05 a**
6	19	6	0	3.26 ± 0.06 b

The bold type indicates the conditions that resulted in the maximum mycelial biomass. Different lowercase letters indicate significant differences between the treatments according to Tukey’s HSD test.

**Table 5 microorganisms-12-02621-t005:** Box–Behnken design structure and experimental mycelial biomass.

Run	Peptone	KH_2_PO_4_	Temperature	Mycelial Biomass
	(%)	(%)	(°C)	(g)
1	0 (5)	0 (0.2)	0 (22)	3.85 ± 0.07 b
2	1 (6)	1 (0)	0 (22)	1.96 ± 0.05 h
3	1 (6)	−1 (0.4)	0 (22)	2.68 ± 0.06 d
4	0 (5)	−1 (0.4)	1 (19)	2.28 ± 0.03 e
5	0 (5)	0 (0.2)	0 (22)	4.15 ± 0.04 a
6	0 (5)	0 (0.2)	0 (22)	3.94 ± 0.03 b
7	1 (6)	0 (0.2)	1 (19)	2.11 ± 0.03 fg
8	0 (5)	0 (0.2)	0 (22)	4.05 ± 0.08 a
9	1 (6)	0 (0.2)	−1 (25)	3.14 ± 0.06 c
10	0 (5)	1 (0)	1 (19)	2.13 ± 0.04 fg
11	0 (5)	−1 (0.4)	−1 (25)	2.58 ± 0.06 d
12	0 (5)	1 (0)	−1 (25)	3.06 ± 0.04 c
13	−1 (4)	1 (0)	0 (22)	2.12 ± 0.06 fg
14	−1 (4)	−1 (0.4)	0 (22)	2.1 ± 0.04 g
15	−1 (4)	0 (0.2)	1 (19)	2.21 ± 0.05 ef
16	−1 (4)	0 (0.2)	−1 (25)	3.11 ± 0.06 c
17	0 (5)	0 (0.2)	0 (22)	3.87 ± 0.04 b

Three factors are shown as real values (coded levels). Mycelial biomass is shown as the mean (standard deviation) of 5 replicates for runs 1–17. The experimental levels for cube (−1 and 1) points were set around the experimental levels of the centre points. Different lowercase letters indicate significant differences between the treatments according to Tukey’s HSD test.

**Table 6 microorganisms-12-02621-t006:** Analysis of variance (ANOVA) for the 8-run Plackett–Burman design and model fitting.

Source	Sum of Squares	df	Mean Square	F-Value	*p*-Value
Model	6.75	8	0.8435	22.39	0.0135 *
A—Corn flour	0.2237	1	0.2237	5.94	0.0928
B—Peptone	0.8397	1	0.8397	22.29	0.018 *
C—MgSO_4_·7H_2_O	0.1764	1	0.1764	4.68	0.1191
D—KH_2_PO_4_	0.6613	1	0.6613	17.55	0.0248 *
E—PH	0.0997	1	0.0997	2.65	0.2023
F—Temperature	0.965	1	0.965	25.61	0.0149 *
G—Filling capacity	3.77	1	3.77	100.13	0.0021 **
H—Rotational speed	0.0093	1	0.0093	0.2462	0.6538
Residual	0.113	3	0.0377		
Cor Total	6.86	11			

R^2^ = 0.9835; R^2^ Adj = 0.9396; R^2^ prediction = 0.7364. ** *p* < 0.01 and * *p* < 0.05.

**Table 7 microorganisms-12-02621-t007:** Regression results of the BBD and ANOVA for the quadratic model.

Source	Sum of Squares	df	Mean Square	F-Value	*p*-Value
Model	10.02	9	1.11	27.14	0.0001
A-Temperature	1.25	1	1.25	30.45	0.0009
B-Peptone	0.0145	1	0.0145	0.3542	0.5705
C-KH_2_PO_4_	0.0162	1	0.0162	0.3955	0.5494
AB	0.0042	1	0.0042	0.1019	0.7589
AC	0.1009	1	0.1009	2.46	0.1608
BC	0.1346	1	0.1346	3.28	0.1131
A^2^	1.12	1	1.12	27.36	0.0012
B^2^	2.79	1	2.79	68.01	<0.0001
C^2^	3.76	1	3.76	91.53	<0.0001
Residual	0.2873	7	0.041		
Lack of Fit	0.2222	3	0.0741	4.56	0.0885
Pure Error	0.065	4	0.0163		
Cor Total	10.31	16			

**Table 8 microorganisms-12-02621-t008:** Distribution of *B. xylophilus* and *M. alternatus* on the pine blocks under different treatments.

Stumps No.	Program	Heights (cm)	Calibre (cm)	Number of Grooves		Number of Feathering Holes		*B. xylophilus* Number		Dry Weight		Mass Loss Percentage (%)
				Before	After	Before	After	Before	After	Before	After	
1	A	15	20	0	1	0	0	608.2 ± 20.75 b	200 ± 19.37 a	2.56 ± 0.06 b	2.16 ± 0.02 b	15.63
2	A	15	15	0	0	1	3	23.4 ± 9.58 e	0 ± 0 c	2.32 ± 0.04 d	1.94 ± 0.07 d	16.72
3	A	25	18	1	3	0	0	311.2 ± 22.95 c	152.4 ± 8.41 b	2.71 ± 0.11 a	2.33 ± 0.01 a	13.94
4	A	36	12	0	1	5	9	589.4 ± 31.6 b	189.6 ± 14.28 ab	2.56 ± 0.08 b	2.11 ± 0.02 b	17.52
5	A	22	18	1	2	2	4	68.2 ± 6.18 e	33.4 ± 5.22 c	2.51 ± 0.09 b	2.09 ± 0.13 b	16.66
6	B	12	15	0	0	0	0	232.4 ± 29.01 d	0 ± 0 c	2.51 ± 0.05 b	1.24 ± 0.03 b	50.81
7	B	15	18	0	0	0	0	669.4 ± 31.5 a	0 ± 0 c	2.37 ± 0.04 cd	1.19 ± 0.04 cd	49.83
8	B	18	23	1	1	1	1	54.4 ± 23.54 e	0 ± 0 c	2.61 ± 0.01 ab	1.33 ± 0.03 ab	48.96
9	B	42	18	2	2	3	3	265 ± 38.96 cd	0 ± 0 c	2.49 ± 0.01 bc	1.21 ± 0.02 bc	51.54
10	B	16	15	0	0	1	1	321.6 ± 36.2 c	0 ± 0 c	2.51 ± 0.04 b	1.25 ± 0.06 b	50.27

In the Program column: A for “non-inoculation” and B for “inoculation”. Different lowercase letters represent significant differences.

**Table 9 microorganisms-12-02621-t009:** Weights of the soil fertility factors.

Target	Cation Exchange Capacity	Total Nitrogen	Total Phosphorus	Total Potassium	Electrical Conductance	Acid Phosphatase
Communality variance	0.708	0.837	0.850	0.893	0.896	0.845
Weight	0.153	0.156	0.182	0.186	0.185	0.138

**Table 10 microorganisms-12-02621-t010:** Membership function turning point of each fertility index.

Turning Point	Cation Exchange Capacity	Total Nitrogen	Total Phosphorus	Total Potassium	Electrical Conductance	Acid Phosphatase
	(cmol·kg^−1^)	(g·kg^−1^)	(g·kg^−1^)	(g·kg^−1^)	(μs·cm^−1^)	(U/g)
X1	10	0.5	0.2	5	120	0.1
X2	20	2	1	25	620	0.54

**Table 11 microorganisms-12-02621-t011:** Fertility affiliation values and composite fertility scores for each soil attribute before and after treatment.

Program	Cation Exchange Capacity	Total Nitrogen	Total Phosphorus	Total Potassium	Electrical Conductance	Acid Phosphatase	Integrated Fertility Index
Pre−treatment	0.065	0.083	0.032	0.176	0.018	0.138	0.512
Non−inoculation	0.060	0.097	0.053	0.184	0.018	0.138	0.551
Inoculation	0.061	0.127	0.054	0.185	0.018	0.138	0.584

## Data Availability

The original contributions presented in this study are included in the article. Further inquiries can be directed to the corresponding author.

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
