# Peer review of "Development of Composite Microbial Products for Managing Pine Wilt Disease in Infected Wood Stumps"

_microorganisms, 2024, doi:10.3390/microorganisms12122621_

Round 1
Reviewer 1 Report
Comments and Suggestions for Authors
In this manuscript, complex studies substantiate the application of microbial products which could be promising for managing pine wilt disease.
The title is complete and matches the content.
The abstract is informative. It presents and summarizes the main results specifically.
The keywords are appropriate.
Introduction. In the introduction, literature sources are analyzed, the problem is formulated and the main goals are presented.
Materials and Methods. The authors indicate how the data were analyzed. It is indicated that “all the experiments were performed in quintuplicate, and the results are presented as the means ± standard deviations”. However, I would suggest specifying exactly which statistical program and its methods were used.
Results. The analysis of the results is presented consistently. On the other hand, only the standard deviations are given in Fig. 4 and 7. The reader would have a clearer assessment of the reliability of the differences if Tukey's test had been applied.
In Table 8, the data are presented without proving the statistical significance of the differences.
In section 3.9, the number of Monochamus alternatus grooves was presented and evaluated. This species appears spontaneously in the Results section, but it is not clear for what purpose. So, it should be substantiated, at least in the Material and Methods section. Serves it as a vector for the nematode Bursaphelenchus xylophilus?
Discussion. In this section, the obtained results were analyzed and compared with the works of other authors, highlighting the novelty of the research presented in this article.
I missed a section on the contributions of authors to this study.
Reviewer 2 Report
Comments and Suggestions for Authors
The manuscript entitled "Development of Composite Microbial Products for Managing Pine Wilt Disease in Infected Wood Stumps" examines the biological control of a significant pine plant protection problem in the world.
The introductory part is detailed enough. The material and method are thoroughly detailed. Allows the reproduction of tests.
In the result chapter white and brown-rot fungi were examined were well known for the ability to break down lignin and cellulose. The combined use of these fungi increased the decomposition of woody substrates. Research has shown that these fungi have an adequate inhibitory effect against the Bursaphelenchus xylophilus nematode, the pine wilt disease (PWD).
In this study, a microbial preparation was studied that effectively breaks down the pine cake while inhibiting the development of B. xylophilus. The correlation between the degradation rate of the wood blocks and the fungal biomass was determined.
In the Conclusion the efficiency and safe use of bioagens were evaluated through field applications in the forest. In addition, the environmental impact of microbial products was evaluated by analyzing soil quality around the treated areas.
The manuscript discusses a sufficient number of up -to -date literature.
After correcting the small letter mistakes, I suggest that the manuscript publish in the form of a scientific article.
Reviewer 3 Report
Comments and Suggestions for Authors
This manuscript presents a comprehensive study on developing composite microbial products for managing pine wilt disease (PWD) in infected wood stumps.
The weakness of this study in my opinion as as follow:
1. The fact that the field tests were conducted in a specific region of China, which may limit the broad application of results to other climatic conditions or pine species.
2. Short-term evaluation or monitoring: Authors of the manuscript didn't mention long-term monitoring of the treated areas, which could be crucial for assessing the lasting effects of the treatment.
Line 63: should be “effective”
Line 101: which plant? do you mean fungi? Please add more details for clarification
Line 136: Please delete the extra blank space it should be 120 μL
Overall, the manuscript presents a well-structured and comprehensive study on developing microbial products for PWD management, with potential for further expansion and refinement in future research.
